



# Characterizing the performance of a POPS
# miniaturized optical particle counter when
# operated on a quadcopter drone
**Zixia Liu[1], Martin Osborne[1,2], Jim Haywood[1,2], Karen Anderson[3],**
**Jamie D. Shutler[3], Andy Wilson[2], Justin Langridge[2], Steve H.L.**
**Yim[4,5,6], Hugh Coe[7], Suresh Babu[8], Sreedharan K. Satheesh[9], Paquita**
**Zuidema[10], Tao Huang[4], Jack C.H. Cheng[4].**
**Affiliations:**
[1]College of Engineering Mathematics, and Physical Sciences, University
of Exeter, Exeter, Devon, UK
[2]Met Office, Fitzroy Road, Exeter, Devon, UK
[3]College of Life and Environmental Sciences, University of Exeter, Penryn,
Cornwall, UK
[4]Department of Geography and Resource Management, The Chinese
University of Hong Kong, Hong Kong, China
[5]Stanley Ho Big Data Decision Analytics Research Centre, The Chinese
University of Hong Kong, Hong Kong, China
[6]Institute of Environment, Energy and Sustainability, The Chinese
University of Hong Kong, Hong Kong, China



[7]Department of Earth and Environmental Sciences, University of
Manchester, Manchester, UK
[8]Space Physics Laboratory, Vikram Sarabhai Space Centre, Trivandrum,
695 022, India
[9]Centre for Atmospheric and Oceanic Sciences, Indian Institute of Science,
Bangalore 560 012, India
[10]Rosenstiel School of Marine and Atmospheric Science, University of
Miami, Miami, FL, USA
*Correspondence to:* J. Haywood (J.M.Haywood@exeter.ac.uk)
**Abstract:**
The Printed Optical Particle Spectrometer (POPS) is an advanced and small low-cost,
light-weight, and high-sensitivity optical particle counter (OPC), particularly designed
for deployed on unpiloted aerial vehicles (UAVs) and balloon sondes. We report the
performance of the POPS against a reference scanning mobility particle sizer (SMPS)
and an airborne passive cavity aerosol spectrometer probe (PCASP) while the POPS is
operated on the ground and also while operated on a quadcopter drone, a DJI Matrice
V2. This is the first such documented test of the performance of a POPS instrument
on a UAV. We investigate the root mean square difference (RMSD) and mean absolute
difference (MAD) in particle number concentrations (PNCs) when operating on the
ground and on the Matrice 200. When windspeeds are less than 2.6m/s, we find only
modest differences in the RMSDs and MADs of 2.4% and 2.3% respectively when
operating on the ground, and to 5% and 3% when operating at 10m altitude. When
windspeeds are greater than 2.6m/s but less than 7.7m/s the RMSDs and MADs
increase to 10.2% and 7.8% respectively when operating on the ground, and 26.2% and




19.1%, respectively when operating at 10m altitude. No statistical difference in PNCs
was detected when operating on the UAV in either ascent or descent. We also find size
distributions of aerosols in the accumulation mode (here defined by diameter, d, where
$0.1 \leq d \leq 1\mu m$) are relatively consistent between measurements at the surface and
measurements at 10m altitude with RMSD and MAD of less than 21.6% and 15.7%,
respectively. However, the differences between coarse mode (here defined by $d > 1\mu m$)
are universally larger than those measured at the surface with a RMSD and MAD
approaching 49.5% and 40.4%. Our results suggest that the impact of the UAV rotors
on the POPS does not unduly affect the performance of the POPS for wind speed less
than 2.6m/s, but when operating under higher wind speed of up to 7.6m/s, larger
discrepancies are noted. In addition to this, it appears that the POPS measures sub-
micron aerosol particles more accurately than super-micron aerosol particles when
airborne on the UAV. These measurements lay the foundations for determining the
magnitude of potential errors that might be introduced into measured aerosol particle
size distributions and concentrations owing to the turbulence created by the rotors on
the UAV.

# 1 Introduction

Atmospheric aerosols have a significant impact on Earth's climate as they affect the
radiative balance of the Earth-Atmosphere system through the direct and indirect effect
(e.g. Haywood and Boucher, 2000; Boucher et al., 2013). The direct effect refers to
absorption and scattering of solar and terrestrial radiation by aerosols, and the indirect
effect refers to the ability of aerosols acting as condensation nuclei (CCN), thereby
influencing cloud microphysical properties (Twomey, 1977), and potentially cloud
extent and lifetime (Albrecht, 1989; Haywood and Boucher, 2000). Although there has
been considerable progress in the knowledge and understanding of aerosol physical and



chemical properties over the past few decades, estimates of aerosol effects of mean
climate impacts and extreme climatic events remain uncertain (Boucher et al., 2013;
Liu et al., 2019a, b). The size, chemical composition, morphology, and number
concentration of aerosols are all important factors in determining their ability to act as
CCN and in their ability to scatter and absorb solar and terrestrial radiation. Aerosol
concentration and their intrinsic properties are spatially inhomogeneous owing to
different emission sources, deposition processes, transports, and chemical reactions (e.g.
Bellouin et al., 2005; Jiminez et al., 2009; Lack and Cappa, 2010; Atkinson et al., 2018;
Yim et al., 2019; Yim et al., 2020). Among these properties, particle size distributions
(PSDs) and number concentrations (PNCs) are of fundamental importance in
determining the impact of aerosols on the atmospheric radiation budget via the aerosol
direct and indirect effects. Based on observations of the size distributions of aerosols
and aerosol refractive index, aerosol optical properties can be inferred (e.g. Atkinson et
al., 2015). The size of aerosol particles is also of primary importance in cloud formation
and precipitation (Yin et al., 2000; Liu et al., 2018; 2019a). As a result, in order to better
understand the effect of aerosols on climate change, it is important to obtain a
comprehensive and accurate characterization of the spatial distribution of aerosol
concentration and properties. Aerosols also play an important role in atmospheric
visibility (e.g. Horvath, 1981), and in respiratory irritants as they are known to have an
adverse impact on air quality and health (e.g. Li et al., 2003; Gu et al., 2016; 2018;
2020; Shi et al., 2019). In terms of scales, satellite observations (e.g. Bellouin et al.,
2005) are able to provide near global coverage of aerosol optical depths, but are only



able to provide bulk measurements of properties of the aerosol size distribution (e.g.
fine mode fraction) and aerosol optical properties (e.g. aerosol absorption). Dedicated
field sites (e.g. Zuidema et al., 2016, 2018) or dedicated sampling with aircraft
instrumentation (e.g. Haywood et al., 2003a; 2020) are able to make much more
detailed aerosol microphysical measurements, but are costly and aircraft cannot sample
aerosols at low altitude in built-up urban regions owing to obvious safety concerns. The
atmospheric science community frequently utilizes optical particle counters (OPCs)
and Mie scattering theory for sizing individual aerosol particles (e.g. Burkhart et al.,
2010). Measurements of aerosols by small, unmanned aerial vehicles (UAVs) have
many advantages, such as low-cost, ease and cost of deployment, and ease of access to
inaccessible areas such as those close to urban conurbations. However, owing to
payload restrictions, UAVs require light-weight, minaturized OPCs. The Printed
Optical Particle Spectrometer (POPS) is an advanced and small low-cost, light-weight,
and high-sensitivity OPC, particularly designed for UAVs and balloon sondes (Gao et
al., 2013; 2016). In brief, the POPS samples particles by drawing air through an inlet
tube into an optical chamber, where it is illuminated by a 405nm laser. A sheath air
flow is used to focus the sample air into the centre of the laser beam, and the sample
flow is maintained at a near constant rate by an automatically regulated on-board pump.
Scattered laser light is reflected into a photomultiplier tube by a hemispherical mirror,
and the signal amplitude recorded by a data logger. Individual particle sizes are then
inferred by comparing the recorded signal amplitudes to scattering amplitudes
calculated using Mie scattering theory.



The POPS has been carried by balloon sondes to study the vertical profile of the Asian
Tropopause Aerosol Layer (Yu et al., 2017), but quantitative data when deployed on a
quadcopter drone is very sparse. There have been some recent side-by-side tests of
miniaturized OPC instruments against more established instrumentation in controlled
environments (e.g. Bezantakos et al., 2018), and some limited comparisons against
large atmospheric tower based instrumentation (Ahn, 2019). A significant question
related to deploying the POPS instrument on a quadcopter drone is whether the
turbulence generated by the multiple rotors impacts the measurements of the aerosol
concentrations and size distributions, and if so, to what extent. Here we provide the first
comprehensive documentation of the performance of the POPS on a multi-rotor UAV.
We first investigated the performance of the POPS instrument in a closely controlled
environment on the ground in a three-week comparison of the POPS against reference
instruments. The POPS was deployed at the Atmospheric Radiation Measurements
(ARM) mobile facility on Ascension Island during colocation of the Layered Atlantic
Smoke Interaction with Clouds (LASIC; Zuidema et al., 2016) and CLoud-Aerosol-
Radiation Interaction and Forcing: Year-2017 (CLARIFY-2017; Haywood et al., 2020)
measurement campaigns. Subsequently we examined the influence of the rotors from
the drone on the measured aerosol number concentration and size distribution. Section
2 presents the methodology used in the ground-based comparison and the UAV-
mounted flights, section 3 presents the results before conclusions and future work is
presented in section 4.



## 2 Methods

### 2.1 A 20-day comparison

As part of the CLARIFY-2017 and LASIC campaign, the POPS was deployed at the ARM mobile site on Ascension Island located in the mid-Atlantic (-7.96° N, -14.37° E) alongside an ARM operated SMPS. The time period for sampling for both instruments analyzed here was from 20th August to 9th September 2017 (20 days) continuously, during which time biomass burning aerosol originating from the African continent was frequently present (Zuidema et al., 2018; Haywood, 2020). The SMPS and the POPS were connected to a common aerosol inlet, however, in the case of the SMPS, the sample air was dried before it entered the instrument.

In common with other OPCs, the POPS size distributions are influenced by the refractive index assumed in the Mie calculations. The manufacturer (Handix Scientific) provides a calibration for the POPS using well-sized latex spheres with a refractive index (RI) of 1.615+0.001i at 405 nm. Prior to deployment to Ascension Island, the manufacturer's calibration of the POPS was adjusted through independent lab-based measurements using latex spheres at the UK's Facility for Airborne Atmospheric Research (FAAM, https://www.faam.ac.uk/). Errors in the PSDs can be caused by sampling aerosols with a different refractive index, particularly if they are significantly absorbing (e.g. Haywood et al., 2003). The independent lab-based calibrations were therefore adjusted assuming a RI of 1.54+0.027i at 405 nm, which is expected to be more representative of the biomass burning aerosol particles sampled at the ARM site





during the CLARIFY deployment (Peers et al., 2019). In the test flights determining
the impact of rotors, the RI of 1.54+0.027i was also applied as it is the relative
difference in the size distribution and concentration that are of most concern during
those operating periods. Compared with the POPS, the SMPS that was operated by the
ARM mobile Facility uses particle mobility subsequent to application of an electrostatic
charge to size aerosol particles, a method which is independent of the refractive index
(Ruzer and Harley, 2012).

In addition to applying fundamentally different methods to measure the size of particles,
the POPS and SMPS cover different ranges of size distributions. The POPS measures
particles within the diameter range from around 0.12 – 4.44μm (for RI = 1.54+0.027i
at 405 nm), while the SMPS covers diameter ranging from around 0.01 to 1.00μm.

## 2.2 Drone-mounted POPS

The POPS required a carefully designed bespoke rig to fit it safely to a quadcopter
drone for deployment. A DJI Matrice 200 V2 was used because it had a sufficient power
and payload capacity to lift the POPS and even with the relatively high payload could
offer reasonable endurance. The maximum flight time of the Matrice is 24 minutes with
the maximum payload (1.45kg). University of Exeter and Met Office staff designed and
fitted the POPS to the Matrice airframe (Figure 1). The POPS was installed at the
bottom left of the fuselage and fixed on the customized 3-D printed landing gear. The
inlet tube of the POPS (red oval in Figure 1) reached 20cm above the rotors. The





diameter of the inlet tube is 1mm and the sample flow rate is 3 cm$^3$/s, yielding a flow
velocity of 3.8m/s. No attempt has been made to optimise this simple tube inlet for
drone applications. The data were collected during 14 test flights in total from 18[th]
December 2019 to 9[th] March 2020 to determine any impact of the rotors on the data
from the POPS. Each test flight was planned to be separated into three stages. During
the first stage, the drone was on the ground with the rotors off for ten minutes (G_NR).
In the second stage, the drone was on the ground with the rotors on for the next ten
minutes (G_R). In the last stage the drone hovered at a fixed position and fixed altitude
of ten meters above the surface for ten minutes (FLY). A summary of date and time of
each test flight is given in Table 1. There are some deviations from the G_NR, G_R and
FLY routines. T1 was a pre-test so there was no FLY. Additionally, due to high wind
speeds and associated operational safety concerns, T9 and T13 had to reduce the test
time of FLY to 7 minutes and 5 minutes, respectively. Three vertical profiles were made
at the end of T10, T12, and T13, the details of which are provided in Table 2. The main
purpose of profiling during T10, T12 and T13 was to investigate: 1) the stability in the
POPS instrument when profiling up and down as this is likely to be a prime operating
maneuver when flying scientific sorties in the future; 2) the performance of the POPS
at different vertical ascent and descent rates; and 3) the accuracy of the POPS on the
way up and way down which could conceivably be influenced by turbulent disturbance
by the rotors, particularly on vertical descents when the aerosol inlet will be in the wake
of the drone rotors. The test flights were all performed at the Streatham campus of the
University of Exeter (50.73N, 3.53W), UK.





# 3 Results

## 3.1 Comparison of POPS data against data from LASIC/CLARIFY-2017.

Figure 2 shows the mean PSD measured by the POPS and SMPS for the 20-day period, respectively. Figure 2 represents the whole size range of the two instruments as well as the fitted PSD from measurements with a wing-mounted PCASP-100X mounted on the UK's Bae146 FAAM aircraft from a flight during CLARIFY-2017 (Peers et al., 2019), which has been shown to be representative of biomass burning aerosol during the wider CLARIFY-2017 measurement campaign (Wu et al., 2020). That the POPS and SMPS show close overlap at the peak concentrations indicates that the counting statistics and the particle concentrations are similar between the instruments. The mean PSD measured by the POPS and SMPS shows reasonable agreement. Although the agreement is not as good as that demonstrated in other comparisons against SMPS instruments (e.g. Gao et al., 2016), any resulting errors in derived optical parameters are likely to be small provided the fit is reasonable over the 0.2-1.0 μm diameter range. Measurements of biomass burning aerosol over the Atlantic from the SAFARI-2000 campaign suggest that particles in this range contribute to 93% of the scattering at 0.55μm (e.g. Table 1, Haywood et al., 2003). The PSD from the wing-borne PCASP-100X that was operated on the FAAM bears a close resemblance to the SMPS and POPS PSDs except at particle sizes <0.2μm diameter and >0.7μm diameter. The discrepancy at particle sizes <0.2μm might be expected because the fits that are adopted



by Haywood et al. (2003) and Peers et al. (2019) do not account for these small particles
as they were developed with simplicity in mind for global general circulation models
of aerosol optical properties and for satellite retrievals respectively. Examination of the
PCASP data to which the log-normal distributions were fitted from both the CLARIFY-
2017 and SAFARI-2000 data indicates higher concentrations of aerosol at these small
sizes than the log-normal fits can represent. Aerosols >0.7μm diameter that were
observed by the POPS that were not present in the CLARIFY-2017 or SAFARI-2000
data may well be generated by dust generation from the arid surface of Ascension Island
or by super-micron sea-salt from breaking waves. Taylor et al (2020) document the
enhanced influence of the oceanic component of aerosols in the marine boundary layer,
but this is not included in the CLARIFY-2017 or SAFARI-2000 log-normal fits which
represent biomass burning aerosols only. Thus, the POPS instrument appears to provide
a reasonably quantitative measure of optically active sub-micron aerosols.

We also investigated the overall particle number concentration from the POPS, and
examine the time series of the POPS measurements against some other key variables
measured by the SMPS and other instrumentation at the ARM mobile facility. The
upper panel of Figure 3a presents the 20-day intercomparison of the PNCs from the
POPS and SMPS, and panel 3b shows the ratio of the two concentration measurements
(POPS/SMPS). They show a good agreement between two instruments while the
geometric mean diameter (GMD) the of the size distribution (Figure 3c) is above





0.12μm. Again, this illustrates that the POPS instrument measures accumulation mode
aerosols reasonably accurately.

We would expect biomass burning aerosols to be associated with an increase in carbon
monoxide (CO; Haywood et al., 2003; Wu et al., 2020), and the concentrations
measured by both the POPS and SMPS instruments are well correlated with the CO
mass mixing ratio (as measured by a co-located CO analyzer – Figure 3d). The
concentration data also show some correlation with the AODs as measured by a co-
located AERONET Cimel sun-photometer (panel five, Figure 3), although this AOD is
a column measurement rather than a point measurement so the influence of vertical
profile will likely be important (e.g. Wu et al, 2020; Haywood et al, 2020).

## 253    3.2 Test flight results

To determine the impact of rotors on the POPS, we focus on the comparison of PSD
and PNC at three different stages: G_NR, G_R, and FLY. Table 3 summaries the mean
PNC with standard deviation and the PNC percentage differences of each flight at
different stages.
3.2.1. Particle number concentration (PNC)
Compared with the mean PNC at G_NR, the mean PNC at G_R changed from -1% to
26%, and that at FLY changed from -1% to 63%, respectively. However, it is apparent
that the differences of PNCs are much lower in the cases T1, T2, T6, T7, and T10 (less
than 10%) in both stages. Figure 4 shows the probability density functions of PNC in



each case. The PNC at three stages of each case were separated into 15 bins. Unpaired
two-sample t-tests were selected to detect the similarity of the PNCs at different stages
as the t-test is the most popular parametric test for samples following normal
distribution for calculating the significance of a small sample size (De Winter, 2013).
Here the PNC of G_NR was set as the control group, while that of G_R and FLY was
set as the perturbation groups using the mean PNC at each stage every 30 seconds.
Before the t-test, the Levene's test was performed which is an inferential statistic used
to assess the equality of variances for a variable calculated for two or more groups
(Levene, 1960). If the Levene's test cannot be passed, then the unequal variances t-test,
which is a more conservative test, was be applied for the groups. The results (p value)
of the t-test of each test flight are shown in Table 4.

For a significance level ($\alpha$) set as 0.05, there are 5 test flights that passed the t test in
both G_R and FLY stages (p value $\geq \alpha$), which means the PNC measured at G_R and
FLY stage corresponded well with those measured at G_NR. These test numbers have
been marked in green and bold italic font in Table 4. This result indicates that the impact
of rotors was not significant in these five cases. The other three cases (T8, T9, and T14)
passed the t-test in the G_R stage, which are marked yellow and italic font. The rest of
test flights did not pass the t-test in either stage (marked red and standard font). Through
comparing the weather conditions, we find that the wind speed (Table 4) was relatively
lower (0.5-2.6m/s) in the cases which pass the t-test at both stages. The wind speed in
the Table 4 was provided from observations at Exeter airport with one-hour resolution.
During the actual experiment, when the wind speed was high, visual observations by



the drone pilot suggested that the drone swung from side to side in the air, causing
increased variability in the pitch, yaw and altitude of the drone. As previously noted,
on T9 and T13 the drone was forced to land early to ensure safety due to the high (>7
m/s) instantaneous wind speed.

To determine the impact of wind speed on PNC observed by the POPS, the cases are
separated into 2 categories: low wind speed (w<2.6m/s) cases and high wind speed
(2.6<w<7.7m/s) cases. The PNC root mean square differences (RMSD) and mean
absolute differences (MAD) at G_R and FLY for all cases, low wind speed cases, and
high wind speed cases are given in the Table 5. For all cases, PNC RMSD is less than
10.2% at G_R and less than 26.2% at FLY, and MAD is less than 7.8% at G_R and less
than 19.1% at FLY. However, in the low wind cases, the RMSD and MAD fall to 2.4%
and 2.3% at G_R, and 5% and 3% at FLY, respectively. In contrast, RMSD and MAD
in the high wind cases increase to 12.6% and 10.9% at G_R, 31.4% and 26.3% at FLY,
respectively. Thus it appears that the inlet air flow of the POPS was not stable when the
drone suffered from variations in pitch and yaw under high wind speed conditions,
which leads to significant fluctuation and variability of the PNC recorded by the POPS
at high wind speed.

3.2.2. The particle size distribution (PSD)
The PSDs at different stages and the mean PSDs ratios at G_R to G_NR and FLY to
G_NR of each test flight are shown in Figure 5, which indicates that the cases with high



similarity of PNCs (T1, T2, T6, T7, and T10) show agreement of the PSD. It also shows
that the differences of sub-micron sizes are less than those of super-micron sizes at G_R
and FLY. Therefore, the size distribution was separated into two modes, the
accumulation mode ($0.1 \leq d \leq 1.0\mu m$) and the coarse mode ($d > 1.0\mu m$), to make a
statistical analysis. Table 6 summaries the PSDs percentage differences for two modes
at G_R and FLY for each case. The PSDs RMSD and MAD for two modes at G_R and
FLY for all cases, low wind cases, and high wind cases are given in Table 6. The
percentage differences of the PSDs are less than 5.4% and 14.9% in low wind cases at
the accumulation mode at G_R and FLY, respectively, while the variation in the PSD in
the coarse mode is perhaps due to lower counting statistics at these sizes. In contrast
PSDs of other cases show differences across the whole spectrum. Even in the
accumulation mode, the differences of the PSDs between FLY and G_NR are up to 53.2%
in the case T8. PSDs RMSD and MAD at the accumulation mode are 3.4% and 2.7%
respectively at G_R in the low wind speed cases, but up to 12.9% and 11.1% at G_R in
the high wind speed cases. These statistics again indicate that impacts of rotors and
UAV attitude on the POPS measurements appear to be reduced in low wind speeds
relative to higher wind speeds. PSD RMSDs and MADs at the coarse mode at G_R,
and at the accumulation and coarse mode at FLY show the same result. Generally
speaking, RMSDs and MADs indicate the impact of rotors and UAV attitude on the
POPS operated in accumulation mode is lower than when in coarse mode, for all cases.
RMSDs in accumulation mode were 10.6% at G_R and 21.6% at FLY, while those in
coarse mode were 32.2% and 49.5% for all cases. MADs showed the same trend as





RMSDs. In the absence of independent multi-stage meteorological tower
measurements (e.g. Ahn, 2019), it is difficult to assess how much of the variability in
PNCs and PSDs is real, particularly when the drone is flying; there may be changes in
PNC with altitude when compared to the surface PNCs owing to surface deposition.
Alternatively, there may be trends in the particle concentrations that occur during the
entire measurement period. A potential solution to the latter would be to change the
three stage sequence from G_NR, G_R, FLY to a five stage sequence of G_NR, and
G_NR. This sequence is suggested for future investigations.

3.2.3. The PNC during vertical profiles
Figure 6 presents the results of the vertical profile runs in T10, T12, and T13. The mean
PNC with standard deviation on the way up and down are shown in Table 7. The PNC
measured on the way up and way down show agreement. The best agreement is found
in the high number concentration, low wind-speed case (T10), where the PNCs differ
by an average of 0.5% between ascent and descent. Even in the high wind-speed cases
when the variability might be expected to be largest owing to changes in the pitch and
yaw of the drone, general agreement is found indicating that the vertical speed of the
drone (which was approximately 0.5 to 1m/s) does not appear to have a significant
impact. Note that the vertical profiles do indicate some variability in the vertical
distribution with PNCs ranging from $1207\pm83$ cm$^{-3}$, $69\pm14$ cm$^{-3}$, and $90\pm11$ cm$^{-3}$ close to
the surface to $1189\pm107$ cm$^{-3}$, $55\pm11$ cm$^{-3}$, and $72\pm15$ cm$^{-3}$ in ascent and $1395\pm83$ cm$^{-3}$,
$69\pm5$ cm$^{-3}$, and $89\pm6$ cm$^{-3}$ close to the surface to $1201\pm101$ cm$^{-3}$, $54\pm12$ cm$^{-3}$, and $82\pm13$
cm$^{-3}$ in descent for flights T10, T12 and T13. This variability with height emphasizes



the utility of small, instrumented UAVs for measuring PNCs and PSDs at low altitudes;
measurements at such altitudes are impossible to probe with heavily equipped
atmospheric research aircraft operating under standard aviation safety protocols.
# 4 Conclusions
We have investigated the performance of POPS against a reference SMPS instrument
while on the ground and also while operated on a quadcopter drone, DJI Matrice 200
V2, which is the first documented test of the performance of a POPS instrument on a
UAV. The investigation includes two parts. The first is a long-term comparison between
the POPS and other instruments during the CLARIFY-2017/LASIC and SAFARI-2000
project. The results show that the PNC measured by the POPS and that measured by
the SMPS and PCASP indicate agreement in the optically important size range centred
at around 0.3μm diameter. This indicates that despite its small size, when operating
under controlled conditions on the ground, the POPS instrument performs relatively
well. In the second part, we tested the impact of drone's rotors and, indirectly the
attitude of the drone, on the performance of the POPS with a focus on two aspects, the
PNC and PSD. We found RMSDs and MADs in PNC when operating a POPS on a
small quad-copter to be less than 10.2% and 7.8%, respectively, when operating on the
ground, and less than 26.2% and 19.1%, respectively, at 10m altitude under wind speed
conditions of up to 7.7m/s. For wind speed of less than 2.6m/s, RMSDs and MADs fell
to 2.4% and 2.3% when operating on the ground, and to 5% and 3% at 10m altitude.
We also found no statistical difference in PNC when operating the UAV in either ascent



or descent. As for the PSD, the accumulation mode aerosol size distributions were
relatively invariant between measurements at the surface and measurements at 10m
altitude with RMSD and MAD of less than 21.6% and 15.7%, respectively. The
differences between coarse mode super-micron aerosols measured at the surface and at
10m altitude were universally greater than those measured at the surface with a RMSD
and MAD approaching 49.5% and 40.4%, but it is unclear whether this is due to loss
of coarse mode aerosol particles to the surface or whether this is due to interference
from the rotors. This impact appears to be most prevalent at the larger end of the POPS
size range. These results suggest that the POPS and UAV and very simple inlet
combination examined here appears able to measure the aerosol PSD and PNC with
reasonable fidelity, particularly for sub-micron aerosols when the wind-speed is
relatively modest.

In follow-up scientific observations, the POPS deployed on the quadcopter drone will
be used to measure the aerosol properties in the atmospheric boundary layer (ABL)
under polluted condition. Concentration of pollutants in the ABL frequently have a
strong correlation with atmospheric stability (Wang et al., 2013, Chambers et al., 2015)
with stable conditions leading to the build-up of pollutants in the ABL. Wind-speeds
are frequently low in stable conditions due to the lack of convection driven turbulence.
Because these future measurements are likely under stable, non-turbulent conditions,
wind-speed effects are not likely to cause significant problems. For other applications
of the POPS on a quadcopter drone, such as the dispersion of pollutants in down-wind





driven plumes, attention should be paid to the influence of the higher wind speeds.
**Acknowledgements:** This work was supported by the Chinese University of Hong
Kong – University of Exeter Joint Centre for ENvironmental SUstainability and
REsilience (ENSURE) programme; ZL, MO, JH, KA, JS, TH, JC and SY would like
to thank ENSURE for their financial support. JH, ZL and HC would also like to
acknowledge NERC SWAAMI (South West Asian Aerosol Monsoon Interactions)
grants NE/L013878/1 and NE/L013886/1 for partial funding of the research.



















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

single scatter albedo and absorption wavelength dependence of black carbon.
Atmospheric Chemistry and Physics, 10, 4207.
Levene, H. 1960. Robust tests for equality of variances. Ingram Olkin, Harold Hotelling, et alia.
Contributions to Probability and Statistics: Essays in Honor of Harold Hotelling.
Stanford University, 278-292.
Li, N. Hao, M. Phalen, R. Hinds, W. and Nel, N. "Particulate air pollutants and asthma: a
paradigm for the role of oxidative stress in PM-induced adverse health effects." Clinical
immunology 109, no. 3 (2003): 250-265.
Liu Z., Ming Y., Zhao C., Lau N.C., Guo J., Bollasina M., Yim S.H.L. (2019a). Contribution of
local and remote anthropogenic aerosols to a record-breaking torrential rainfall event
in Guangdong Province, China. Atmospheric Chemistry and Physics, 20, 223-241.
Liu Z., Ming Y., Wang L., Bollasina M., Luo M., Lau N.C., Yim S.H.L. (2019b). A Model
Investigation of Aerosol-Induced Changes in the East Asian Winter Monsoon.
Geophysical Research Letters, 46(16), 10186-10195.





Liu Z., Yim S.H.L., Wang C., Lau N.C. (2018). The impact of the aerosol direct radiative forcing
on deep convection and air quality in the Pearl River Delta region. *Geophysical*
*Research Letters*, 45(9), 4410-4418.
Peers, F., Francis, P., Fox, C., Abel, S.J., Szpek, K., Cotterell, M.I., Davies, N.W., Langridge,
509         J.M., Meyer, K.G., Platnick, S.E. and Haywood, J.M., 2019. Observation of absorbing
aerosols above clouds over the south-east Atlantic Ocean from the geostationary
satellite SEVIRI–Part 1: Method description and sensitivity. *Atmospheric Chemistry and*
*Physics*, 19, 9595–9611, 2019 https://doi.org/10.5194/acp-19-9595-2019.
Ruzer, L. S. & Harley, N. H. 2012. Aerosols handbook: measurement, dosimetry, and health
effects, CRC press.
Sang-Nourpour, N., & Olfert, J. S. (2019). Calibration of optical particle counters with an
aerodynamic aerosol classifier. *Journal of Aerosol Science*, 138, 105452.
Shi C., Nduka I.C., Yang Y., Huang Y., Yao R., Zhang H., He B., Xie C., Wang Z., Yim S.H.L.
(2019). Characteristics and Meteorological Mechanisms of Transboundary Air Pollution
in A Persistent Heavy PM2.5 Pollution Episode in Central-East China. *Atmospheric*
*Environment*, 223, 117239.
Taylor, J. W., Wu, H., Szpek, K., Bower, K., Crawford, I., Flynn, M. J., Williams, P. I., Dorsey, J.,
Langridge, J. M., Cotterell, M. I., Fox, C., Davies, N. W., Haywood, J. M., and Coe, H.:
Absorption closure in highly aged biomass burning smoke, *Atmospheric Chemistry and*
*Physics*, 20, 11201–11221, https://doi.org/10.5194/acp-20-11201-2020, 2020.
Twomey, S. (1977). The influence of pollution on the shortwave albedo of clouds. *Journal of the*
*atmospheric sciences*, 34(7), 1149-1152.
Wang, F., Zang, H., Ancora, M. & Deng, X. 2013. Measurement of atmospheric stability index
by monitoring radon natural radioactivity. *China Environmental Science*, 33, 594-598.
Wu, H.,   J.W. Taylor, K. Szpek, P.I. Williams, M. Flynn, J. Langridge, J.D. Allan, J. Pitt, S. Abel,
530         J. Haywood, H. Coe, Vertical and temporal variability of the properties of transported
biomass burning aerosol over the southeast Atlantic during CLARIFY-2017, *ACPD.*.
Yim S.H.L., Gu Y., Shapiro M., Stephens B. (2019). Air quality and acid deposition impacts of
local emissions and transboundary air pollution in Japan and South Korea. *Atmospheric*
*Chemistry and Physics*, 19, 13309-13323.
Yim S.H.L. (2020). Development of a 3D Real-Time Atmospheric Monitoring System
(3DREAMS) Using Doppler LiDARs and Applications for Long-Term Analysis and Hot-
and-Polluted Episodes. *Remote Sensing* 12 (6), 1036.
Yin, Y., Levin, Z., Reisin, T. G. & Tzivion, S. 2000. The effects of giant cloud condensation nuclei
on the development of precipitation in convective clouds—A numerical study.
*Atmospheric Research*, 53, 91-116.
Yu, P., Rosenlof, K. H., Liu, S., Telg, H., Thornberry, T. D., Rollins, A. W., Portmann, R. W., Bai,
Z., Ray, E. A. & Duan, Y. 2017. Efficient transport of tropospheric aerosol into the
stratosphere via the Asian summer monsoon anticyclone. *Proceedings of the National*
*Academy of Sciences*, 114, 6972-6977.
Zuidema, P., Sedlacek III, A.J., Flynn, C., Springston, S., Delgadillo, R., Zhang, J., Aiken, A.C.,
Koontz, A. and Muradyan, P., 2018. The Ascension Island boundary layer in the remote
southeast Atlantic is often smoky. *Geophysical Research Letters*, 45(9), pp.4456-4465.
Zuidema, P., Redemann, J., Haywood, J., Wood, R., Piketh, S., Hipondoka, M. and Formenti,





P., 2016. Smoke and clouds above the southeast Atlantic: Upcoming field campaigns
probe absorbing aerosol's impact on climate. *Bulletin of the American Meteorological*
*Society*, 97(7), pp.1131-1135.



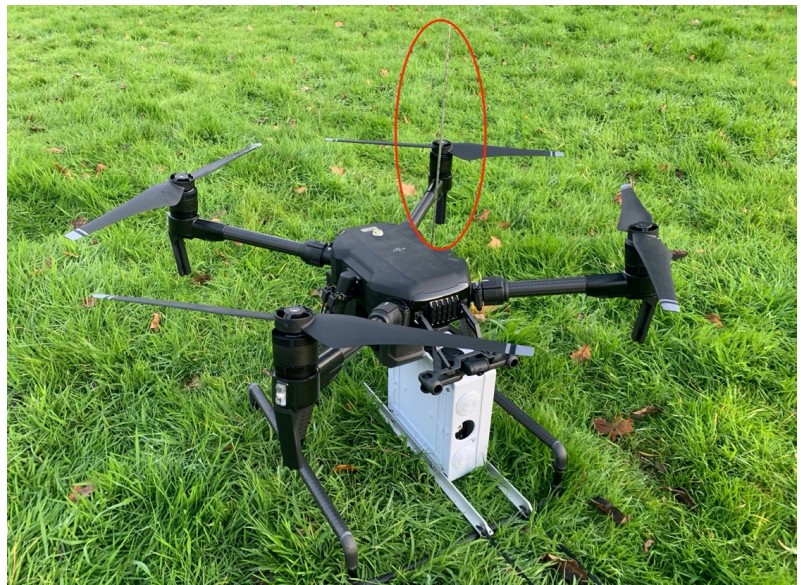


**Figure 1.** The DJI Matrice 200 V2 with the POPS (white box at the left bottom of the fuselage).
The red oval shows the inlet tube leading to the POPS.




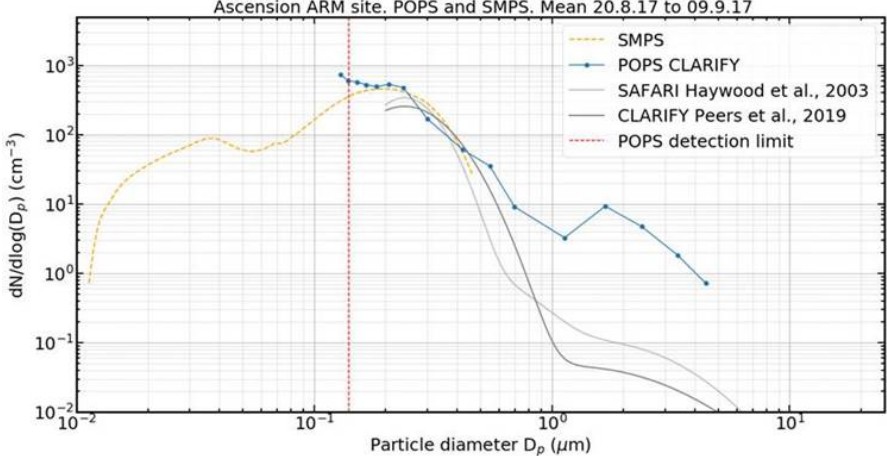


**Figure 2.** PSDs from POPS, SMPS, and data fitted to a wing-mounted PCASP from CLARIFY-
2017 and SAFARI-2000. POPS and SMPS data were collected at the ARM mobile site on Ascension
Island from 20[th] of August to 9[th] of September 2017.The PCASP data from CLARIFY were
collected from a flight on 4[th] of September 2017 (Peers et al., 2019). The PCASP data from SAFARI-
2000 represent a mean from 11 flights performed off the coast of Namibia (Haywood et al., 2003).
Note that the CLARIFY-210 and SAFARAI-2000 PCASP distributions are 'scaled' to the SMPS
size distribution to aid comparison. The POPS and SMPS values are not scaled.



607

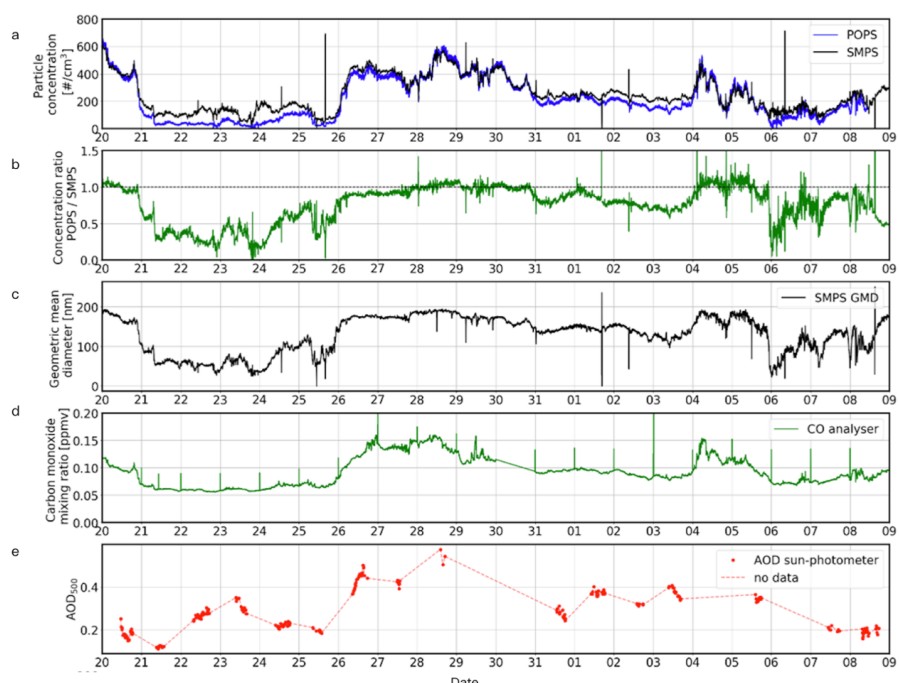

608

**Figure 3.** From top to bottom. (a) SMPS and POPS total particle concentration. (b) Ratio of POPS to SMPS total particle concentration. (c) Geometric mean diameter from SMPS. (d) Carbon monoxide mixing ratio from Los Gatos Research CO analyser, and (e) AOD from Cimel sun-photometer.


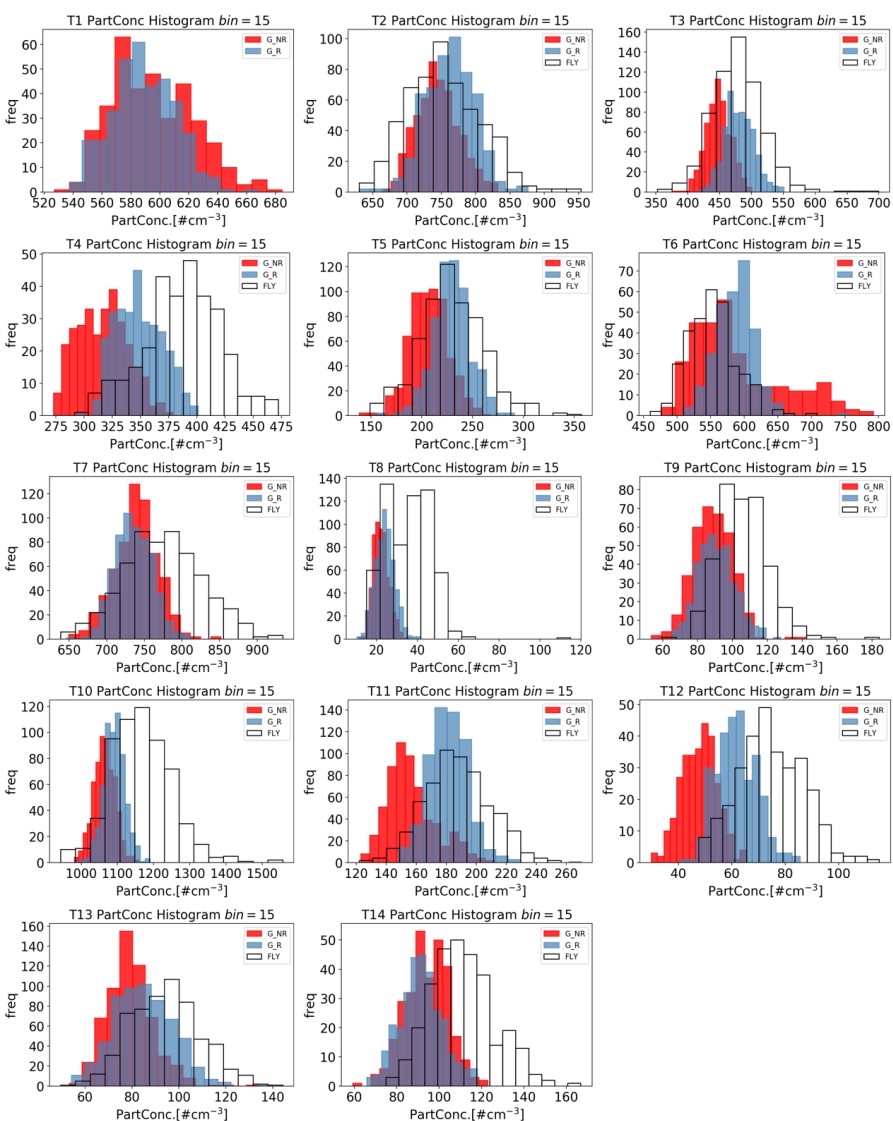


**Figure 4.** Probability density functions of PNCs in each case. The bin number was set to 15 for all
stages. Red represents the G_NR, blue represents G_R, and white represents FLY.












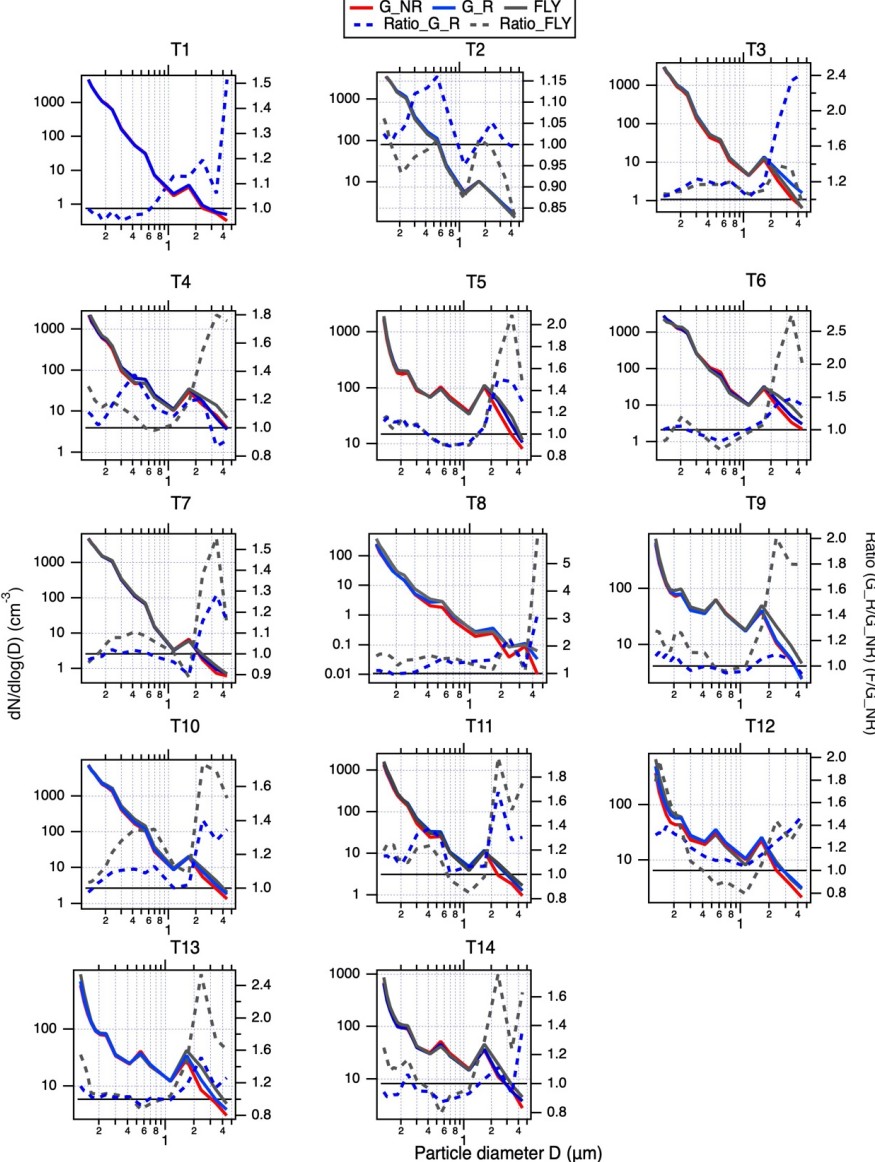


**Figure 5.** Particle size distribution at three stages: the drone on the ground with rotors off (G_NR) (red line), on the ground with rotors on (G_R) (blue line) and flying at 10m (FLY) (grey line), in each POPS test. The ratios of the PSD at G_R to G_NR (blue dash line) and at FLY to G_NR (grey dash line) of each flight are given in each plot.

648

649

650

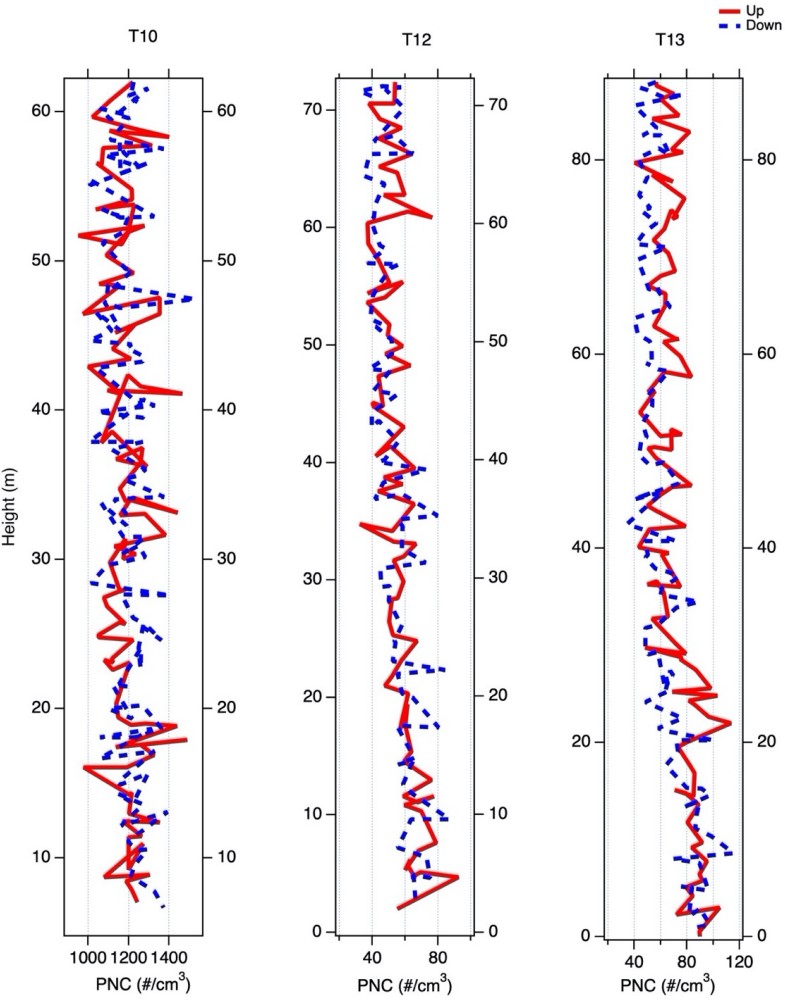

**Figure 6.** Vertical profiles of the particle number concentration in the profile runs of T10, T12, and T13. The red line shows the observed concentration in the way up and the blue dash line shows that in the way down, respectively.





|  | Date | Time |
|---|---|---|
| T1 | 18/Nov/2019 | 16:07 - 16:30 |
| T2 | 19/Nov/2019 | 17:00 - 17:35 |
| T3 | 20/Nov/2019 | 14:20 - 15:10 |
| T4 | 25/Nov/2019 | 10:36 - 11:15 |
| T5 | 26/Nov/2019 | 15:21 - 16:00 |
| T6 | 28/Nov/2019 | 11:08 - 11:46 |
| T7 | 2/Dec/2019 | 11:45 – 12:31 |
| T8 | 30/Jan/2020 | 11:49 – 12:34 |
| T9 | 4/Feb/2020 | 10:41 – 11:15 |
| T10 | 7/Feb/2020 | 11:57 – 12:44 |
| T11 | 12/Feb/2020 | 16:35 – 17:26 |
| T12 | 26/Feb/2020 | 14:36 – 17:27 |
| T13 | 3/Mar/2020 | 11:24 – 12:06 |
| T14 | 9/March/2020 | 11:55 – 12:28 |

**Table 1.** Summary of the dates and time of each test flight.

|  | Initial Height (m) | End Height (m) | Vertical Speed (m/s) |
|---|---|---|---|
| T10 | 5 | 60 | 0.5 |
| T12 | 5 | 70 | 1 |
| T13 | 2 | 90 | 1 |

**Table 2.** Summary of the initial and end heights and vertical speed of each vertical profile.






| | Date | Particle Number Concentrations (PNCs) (cm$^{-3}$) | | | Percentage Difference (%) | |
|---|---|---|---|---|---|---|
| | | G_NR | G_R | FLY | G_R | FLY |
| T1 | 18/Nov/2019 | 597±30 | 587±22 | n/a | -1.7 | n/a |
| T2 | 19/Nov/2019 | 741±52 | 767±35 | 742±31 | 3.5 | 0.1 |
| T3 | 20/Nov/2019 | 442±48 | 479±23 | 478±40 | 8.4 | 8.1 |
| T4 | 25/Nov/2019 | 317±36 | 349±21 | 385±30 | 10.1 | 21.5 |
| T5 | 26/Nov/2019 | 207±19 | 228±18 | 230±31 | 10.1 | 11.1 |
| T6 | 28/Nov/2019 | 567±50 | 580±30 | 561±41 | 2.3 | -1.1 |
| T7 | 2/Dec/2019 | 753±30 | 745±24 | 760±55 | -1.1 | 0.9 |
| T8 | 30/Jan/2020 | 22±4 | 24±5 | 36±11 | 9.1 | 63.6 |
| T9 | 4/Feb/2020 | 87±11 | 91±11 | 105±19 | 4.6 | 20.7 |
| T10 | 7/Feb/2020 | 1063±29 | 1092±29 | 1169±84 | 2.7 | 9.9 |
| T11 | 12/Feb/2020 | 156±16 | 181±13 | 187±21 | 16.0 | 19.9 |
| T12 | 26/Feb/2020 | 50±7 | 63±9 | 74±11 | 26 | 48 |
| T13 | 3/Mar/2020 | 79±10 | 86±13 | 102±13 | 8.9 | 29.1 |
| T14 | 9/March/2020 | 95±12 | 90±10 | 108±14 | -5.3 | 13.7 |


**Table 3.** Summary of the PNCs of each test flight at three stages. n/a = not applicable. The numbers
denoted by ±x represent the standard deviation in the PNCs during the measurement time period.



























| | Surface Wind Speed (m/s) | T test P value | |
|---|---|---|---|
| | | G_R | FLY |
| *T1* | 0.5 | 0.2 | n/a |
| *T2* | 2.6 | 0.3 | 0.6 |
| T3 | 5.7 | 2e-9 | 2e-7 |
| T4 | 3.6 | 8e-5 | 2e-7 |
| T5 | 6.7 | 2e-9 | 2e-6 |
| *T6* | 1.5 | 0.9 | 0.2 |
| *T7* | 1 | 0.9 | 0.3 |
| *T8* | 4.1 | 0.05 | 3e-6 |
| *T9* | 7.7 | 0.2 | 1e-10 |
| *T10* | n/a | 0.7 | 0.2 |
| *T11* | n/a | 2e-5 | 5e-6 |
| *T12* | n/a | 4e-10 | 1e-6 |
| *T13* | n/a | 0.02 | 1e-14 |
| T14 | n/a | 0.2 | 1e-5 |

**Table 4.** Summary of the dates, time, wind speed, and t test results (p value) of each test flight. Wind speed values (at 1.5m) are the wind speed in the hour closest to the experiment time. From T10 to T14 the wind speed data is not available (n/a) because the instrument recording the data had broken. Flights highlighted in green and bold italic font indicate that the results are not significantly different at 5% significance. Flights marked in yellow and italic font indicate that the PNC on the ground with the rotor on are not significantly different from G_NR, and flights marked in red and standard font indicate that there are significant differences in both G_R and FLY when compared to G_NR.

| | PNC RMSD (%) | |
|---|---|---|
| | G_R | FLY |
| All cases | 10.2 | 26.2 |
| Low wind speed cases (w<2.6m/s) | 2.4 | 5 |
| High wind speed cases (2.6<w<7.7m/s) | 12.6 | 31.4 |
| | PNC MAD (%) | |
| | G_R | FLY |
| All flights | 7.8 | 19.1 |
| Low wind speed cases (w<2.6m/s) | 2.3 | 3 |
| High wind speed cases (2.6<w<7.7m/s) | 10.9 | 26.3 |

**Table 5.** Summary of RMSD and MAD for all cases, low wind cases, and high wind cases.



| | Mean Percentage Difference (%) | | | |
| --- | --- | --- | --- | --- |
| | G_R (%) | | FLY (%) | |
| | Accumulation | Coarse | Accumulation | Coarse |
| *T1* | -2.1 | 17.5 | n/a | n/a |
| *T2* | 5.4 | 1.6 | -0.7 | -7.0 |
| T3 | 10.8 | 67.4 | 9.7 | 17.5 |
| T4 | 13.3 | 6.8 | 15.0 | 38.3 |
| T5 | 7.7 | 19.5 | 6.4 | 35.6 |
| *T6* | -0.8 | 22.0 | -3.6 | 61.8 |
| *T7* | -0.3 | 7.5 | 3.3 | 17.2 |
| *T8* | 11.6 | 83.0 | 53.2 | 123.1 |
| *T9* | 4.2 | 0.9 | 15.6 | 48.0 |
| *T10* | 4.9 | 19.8 | 14.9 | 42.4 |
| *T11* | 18.0 | 23.8 | 16.8 | 33.9 |
| *T12* | 25.2 | 23.0 | 43.2 | 14.8 |
| *T13* | 4.2 | 18.4 | 13.9 | 55.5 |
| T14 | -5.1 | 4.5 | 7.3 | 29.9 |
| | RMSD (%) | | | |
| | G_R | | FLY | |
| | Accumulation | Coarse | Accumulation | Coarse |
| All cases | 10.6 | 32.2 | 21.6 | 49.5 |
| Low wind speed cases (w<2.6m/s) | 3.4 | 15.8 | 7.8 | 38.6 |
| High wind speed cases (2.6<w<7.7m/s) | 12.9 | 38.5 | 25.4 | 53.6 |
| | MAD (%) | | | |
| | G_R | | FLY | |
| | Accumulation | Coarse | Accumulation | Coarse |
| All cases | 8.1 | 22.6 | 15.7 | 40.4 |
| Low wind speed cases (w<2.6m/s) | 2.7 | 13.7 | 5.6 | 32.1 |
| High wind speed cases (2.6<w<7.7m/s) | 11.1 | 27.5 | 20.1 | 44.1 |

**Table 6.** Summary of mean percentage differences of size distribution between G_NR and G_R,
and G_NR and FLY of each flight. The size distributions are separated into two modes:
accumulation mode ($0.1 \leq d \leq 1\mu m$) and coarse mode ($d > 1\mu m$).



| | Mean PNCs ($cm^{-3}$) | |
|---|---|---|
| | Up | Down |
| T10 | 1189±107 | 1201±101 |
| T12 | 55±11 | 54±12 |
| T13 | 72±15 | 82±13 |

**Table 7.** Mean PNC with standard deviations on the way up and down in three vertical profile
runs.