# Peer review of "Characterizing the performance of a POPS"

_Atmospheric Measurement Techniques, 2020_

## Author Comment (AC1)

**Characterizing the performance of a POPS miniaturized optical particle counter when operated on a quadcopter drone, by Liu et al.**

**Response to Reviewers**

We would like to thank the reviewers for their comments on the manuscript. The reviewers raised some interesting and relevant points that we have addressed in our revised manuscript. Our responses are presented in red.

**First Referee**

**Overall Comments:**

The authors present a ground-based comparison of a POPS measurement and SMPS in an environment heavily influenced by biomass burning aerosol. The authors then compare POPS measurements on a UAS in flight to its measurements on the ground and suggest that these differences are due primarily to the UAS flight.

Although comparing an SMPS to the POPS is not particularly novel, this study does verify that the agreement holds in an environment influenced by biomass burning and where a different index of refraction has been assumed.

The authors suggest that the POPS is adversely affected in this particularly sampling position on a small rotary UAS. Yet the authors note that no attempts were made to sample from other locations on the multi-rotor aircraft, or to modify the inlet for in-flight aerosol measurements. Potential perturbation to instrument flow are not directly addressed. These should perhaps be persued. At the very least, the authors must tailor the scope of their claim to say that simplistic measurements form the POPS instrument are ill advised and quad copter aerosol sampling requires careful evaluation and consideration.

One of the questions that is most frequently asked when sampling aerosols on large state-of-the-art atmospheric measurement aircraft (e.g. the UK's FAAM platform) is whether the inlets and attitude of the aircraft influence microphysical measurements of aerosols – particularly the aerosol number distribution and the derived size distribution. This is not a new problem and has been recognised for decades (e.g. Huebert et al., 1990). The same questions apply, but for different reasons, to quadcopter UAV based sampling of aerosols because there four rotors in close proximity to the sampling inlet and the platform is necessarily not as stable as a large airborne platforms leading to greater variability in the attitude of the sampling platform caused by auto-corrections to maintain the UAV's position. The purpose of the measurements is to quantify any potential error that might occur owing to these factors even from a relatively simplistic measurement approach. We concentrate on assessing this influence and suggest that the errors are relatively minor at low-wind speeds. We report the

results objectively with statistical analysis at every point and report the following in the (revised) abstract that the reviewer asked to be shortened abstract:-

"When windspeeds are low (less than 2.6 m/s), we find only modest differences in the RMSDs and MADs of 5% and 3% when operating at 10m altitude."

"No statistical difference in PNCs was detected when operating on the UAV in either ascent or descent."

"These measurements lay the foundations for determining the magnitude of potential errors that might be introduced into measured aerosol particle size distributions and concentrations owing to the turbulence created by the rotors on the UAV."

We therefore refute the reviewer's comments that when operating on a UAV "measurements form the POPS instrument are ill advised". They may be ill-advised under high wind-speeds, but not under light windspeeds. If the reviewer can point us to other studies that have shown that measurements with lightweight OPCs are universally ill-advised, then we would appreciate them directing us to the relevant studies.

We clearly state the objectives in the introduction:-

"A similar significant question related to deploying the POPS instrument on a quadcopter drone is whether the turbulence generated by the multiple rotors impacts the measurements of the aerosol concentrations and size distributions, and if so, to what extent. Here we provide the first comprehensive documentation of the performance of the POPS on a multi-rotor UAV."

Given the reviewer's comments, we modify this text to:

"Questions about the impact of inlets, and aircraft boundary layer depths on aerosol measurements have been the subject of research on aerosol for decades (Huebert et al., 1990; Sanchez-Marroquin, 2019). A similar significant question related to deploying the POPS instrument on a quadcopter drone is whether the turbulence generated by the multiple rotors and the attitude adjustment required to maintain positional stability impact the measurements of the aerosol concentrations and size distributions, and if so, to what extent. Here we provide the first comprehensive documentation of the performance of the POPS on a multi-rotor UAV."

Huebert, B. J., Lee, G., & Warren, W. L. (1990). Airborne aerosol inlet passing efficiency measurement. Journal of Geophysical Research: Atmospheres, 95(D10), 16369-16381.

Sanchez-Marroquin, A., Hedges, D. H. P., Hiscock, M., Parker, S. T., Rosenberg, P. D., Trembath, J., Walshaw, R., Burke, I. T., McQuaid, J. B., and Murray, B. J.: Characterisation of the filter inlet system on the FAAM BAe-146 research aircraft and its use for size-resolved aerosol composition measurements, Atmos. Meas. Tech., 12, 5741–5763, https://doi.org/10.5194/amt-12-5741-2019, 2019.

The flight tests do not involve an in-flight inter-comparison, which we suggest the authors also pursue, if possible. As noted, real differences in aerosol distributions and particle concentration number with time could have obscured perceived UAS aerosol sampling bias.

It is not possible at present to pursue an in-flight comparison. Using a large airborne platform down to 10m altitudes would obviously bring its own problems. Measurements could perhaps be made using missed approaches, but the spatial scales sampled by e.g. the FAAM aircraft travelling at 100m/s would bring as many questions to the table as they answer as you cannot operate UAVs at airfields. It might be possible in the future to make measurements using a large atmospheric tower. However, multiple POPS instruments would be required, one at each measurement altitude and one at the surface. This is frankly beyond the financial scope of the current project.

Again, we have already highlighted (the very limited number of) previous studies (lines 101-107):-
"There have been some recent side-by-side tests of miniaturized OPC instruments against more established instrumentation in controlled environments. For example, Bezantakos et al. (2018) compared a newly developed miniaturized OPC against a GRIMM OPC across a range of atmospheric conditions. There have also been some very limited comparisons of miniaturized UAV-borne OPC instrumentation against measurements on large atmospheric tower based instrumentation (Ahn, 2019). Neither of these studies use the POPS OPC.". Note also that the Ahn reference is 'grey' literature from the 7th International Symposium on Ultrafine Particles – this simply reflects that studies such as that documented here are very, very scarce and the subject is novel.

We are very frank that we could potentially have made additional measurements on the surface subsequent to the flight (lines 343-345) and that this should guide future research efforts:
However, given the number of flights that we have performed with three stages (thirteen as detailed in Table 3), we have performed further analysis that shows that it is close to inconceivable that significant trends in the same direction would occur in ALL of the flights over a duration of around 30minutes (10 minutes on the ground, 10 minutes on the ground with rotors on, 10 minutes flying at 10m altitude). To back-up this assertion we analyse any trends in the G_NR and G_R statistics by determining the mean slope (particles / s) during the operating periods; only flights T4 and T6 show trends of greater than 0.1 particles per second

when averaged over both G_NR and G_R. Figure 4 shows that there is potentially an increase in the concentrations that are measured during T4, but that there is potentially a much broader and potentially bi-modal number concentration measured during T6. Thus it can be inferred that there is no evidence of a significant trend in atmospheric concentrations across all flights.

We therefore changes our statement to "Alternatively, there may be trends in the particle concentrations that occur during the entire measurement period which spanned around 30 minutes duration. We determine trends in the G_NR and G_R statistics by determining the mean slope (particles / s) during the operating periods; only flights T4 and T6 show trends of greater than 0.1 particles per second (6 particles / minute) when averaged over both G_NR and G_R. Figure 4 shows that there is potentially an increase in the concentrations that are measured during T4, and that there is potentially a bi-modal number concentration measured during the G_NR sampling period for T6. As no trends are evident for the other flights, it can be inferred that there is no evidence of a systematic significant trend in atmospheric concentrations across all flights; any such trends are likely to be random. However, a potential solution to any concern would be to change the three stage sequence from G_NR, G_R, FLY to a five stage sequence of G_NR, G_R, FLY, G_R and G_NR. This sequence is suggested for future investigations."

**Specific Comments:**

The abstract should be shortened.

We have shortened the abstract to 291 words. However, we note that this is really a matter of personal preference and is not inconsistent with other articles recently published in AMT:-

Xu, N. and Collins, D. R.: Design and characterization of a new oxidation flow reactor for laboratory and long-term ambient studies, Atmos. Meas. Tech., 14, 2891–2906, https://doi.org/10.5194/amt-14-2891-2021, 2021. (398 words)

L31 – L33 An odd comment; remove from the abstract. Begin with, "we compared the

Portable Optical Particle Spectrometer, a small light-weight and high sensitivity optical

particle counter…"

We change our statement to "We first validate the performance of the Portable Optical Particle Spectrometer, a small light-weight and high sensitivity optical particle counter, against a reference scanning mobility particle sizer (SMPS) for a month-long deployment in an environment dominated by biomass burning aerosols. Subsequently, we examine any biases introduced by operating the POPS on a quadcopter drone, a DJI Matrice 200 V2."

L37 Awkward. Rephrase. "This is the first such documented…

Removed.

L38 Word choice – you don't "investigate" the RMSE or MAD - you report it.

We modified the text to "We report the root mean square difference (RMSD) and mean absolute difference (MAD) in particle number concentrations (PNCs) when operating on the ground and on the drone."

L50 – 52 Be specific about differences in coarse mode to what other instrument – SMPS

does not measure particles > 1.0 micron. This is particularly confusing…

For this part, the differences are not compared with SMPS. We hope that the abstract now reflects the separation between the long-term observation in the biomass burning rich environment (Ascension Island) and the measurements performed in the UK testing the influence of mounting the instrument on a UAV.

L62-74 This section could be much more concise.

We modified the text to:

"Atmospheric aerosols have a significant impact on Earth's climate as they affect the radiative balance of the Earth-Atmosphere system through the direct effect which refers to absorption and scattering of solar and terrestrial radiation, and the indirect effect which refers to the ability of aerosols acting as condensation nuclei (CCN) (Haywood and Boucher, 2000; Boucher et al., 2013)."

L88 Awkward phrasing.

We rephrase the sentence to: "Aerosols can also impact atmospheric visibility (e.g. Horvath, 1981), air quality, and health (e.g. Li et al., 2003; Gu et al., 2016; 2018; 2020; Shi et al., 2019)."

L106 – 113 This section could be shortened and only details particularly relevant to this

study should be mentioned (this overall description is covered in Gao et al. 2013 and

2016).

We prefer to keep this section because it provides a better description of the POPS for readers. It is already very short.

L117 Did this study include a POPS?

See below

L119 Comparisons to tower measurements – what instruments were compared and were

they compared only at one height? Was temporal averaging applied?

At the request of the reviewer, we now include a little more detail:

"There have been some recent side-by-side tests of miniaturized OPC instruments against more established instrumentation in controlled environments. For example, Bezantakos et al. (2018) compared a newly developed miniaturized OPC against a GRIMM OPC across a range of atmospheric conditions. There have also been some very limited comparisons of miniaturized UAV-borne OPC instrumentation against measurements on large atmospheric tower based instrumentation (Ahn, 2019). Neither of these studies use the POPS OPC."

L131-132 It is still not clear what was entailed in the in-flight UAS POPS comparison. Please be more specific.

We think that the reviewer is confused between the two aspects of the study: an intercomparison against established instrumentation when operating on the ground (Ascension Island) followed by an investigation of whether any biases were introduced when operating the instrument on a drone (UK).

L132-134. This sentence should be removed. It is not helpful.

We disagree. This provides the roadmap of what the reader is to expect: Section 2 provides details of the methods, section 3 the results before conclusions and future work are presented in section 4.

L153-155 Was the adjustment to account for a difference in the index of refraction done to

binned data or per particle data? Doing this to binned data could introduce an additional

(likely small) source of error.

The adjustment to account for a difference in the index of refraction was performed on binned data. To do otherwise makes no sense as the biomass burning aerosol data measured at the Ascension Island site will have variable refractive indices on a particle by particle basis. This is not a controlled environment. We therefore make a small adjustment to the sentence:

"The independent lab-based calibration binning criteria were therefore adjusted assuming a RI of 1.54+0.027i"

L178-180 Can the authors comment on how the sampling tube might be optimized for

drone sampling? This seems like a very important point considering the comparison/ test.

As discussed at length in our first few paragraphs, the important point of this study is not to provide an improvement of the inlet, but to estimate the performance of the POPS on a quadcopter drone with a simple inlet. What error does this induce? How does it depend on wind-speed and how that might upset the attitude of the drone? Are there differences in the measurements when the UAV is going up (and the inlet is ahead of the wake from the rotors) versus going down (when the inlet could be more likely affected by the wake from the rotors)?

L205 Was this date of the wing-mounted PCASP instrument a day that the POPS sampled

(on the UAS or on the ground)? What altitudes were sampled to provide these size

distributions?

If not, perhaps shorten this section and specify that the PCASP size distribution is simply

provided for reference.

The PCASP data are from two different projects, one is from the SAFARI (measurements made in 2000), another one is from the CLARIFY (measurements made in 2017).

The POPS data are from measurements on Ascension Island during the CLARIFY project. It was collected from the ARM mobile site on Ascension Island from 20th August to 9th September 2017 at around 330m altitude.

PCASP data from the CLARIFY campaign was collected on 4th September 2017 in the vicinity of Ascension Island but from 7.3 to 1.9 km altitude. Aged biomass burning size distributions have been shown to be quite invariant over the entire SE Atlantic region during the biomass burning season (e.g. Yu et al., 2020; Taylor et al., 2020). The PCASP data from the SAFARI was collected in 2000 off the coast of Namibia and shows a striking resemblance to that collected during the CLARIFY program. The PCASP is an optical measurement while the SMPS is a mobility measurement. So one might on the face-of-it expect a better agreement between the PCASP and the POPS given the invariance in the observed aerosol size distribution around Ascension Island and the similarity in the OPC measurement methods. That the POPS measurements agree with both the SMPS and the PCASP measurements gives evidence that the POPS instrument is doing a reasonable job.

However, at the request of the reviewer we shorten this section as per the amendments in the revised paper.

L235-240 For an instrument comparison, the size range of the SMPS and POPS should

only be compared in the range where measurements overlap. The full SMPS and POPS size

ranges should only be used to characterize atmospheric aerosol distributions more fully.

This is a good point. We have done as the reviewer suggested and revised the first two panels of the figure including just the overlapped number concentration. The differences are relatively minor owing to the peak in the size distribution occurring in the accumulation mode. We have adjusted the figure caption to reflect that we are comparing the concentration from the bins that overlap only.

[Figure]

**Figure 3.** From top to bottom. (a) SMPS and POPS total particle concentration. The number distribution is calculated over the range of bins that overlap between the two instruments (approximately 120 - 450nm diameter). (b) Ratio of POPS to SMPS total particle concentration derived from (a). (c) Geometric mean diameter from SMPS. (d) Carbon monoxide mixing ratio from Los Gatos Research CO analyser, and (e) AOD from Cimel sun- photometer. Spikes in the CO data occur at the beginning of each day when the instrument is in calibration mode.

L258 – 290. Since no in-flight comparison to another instrument was done, the authors

need to demonstrate that they did not observe any systematic differences in the PSD at

10 m compared with at the ground.

Agreed. Table 3 and Table 4 show that the differences when the wind-speed is low are small in terms of the number distribution measured by the POPS (flights 1, 2,6,7 and 10) – look at the PNC differences between the FLY and G_NR. They are : n/a, 0.1%, -1.1%, 0.9% and 9.9%. These differences in PNC are further analysed by determining the P-values: n/a, 0.6, 0.2, 0.3, 0.2. These are all indicative that the PNCs at 10m and at the surface with the rotors off are statistically identical. Of course, the PNCs are going to be dominated by the more numerous small particles. We then go on to examine whether there are difference systematic differences in the size distributions at 10m and at the ground. We do find differences and we report them.

As we document earlier in the responses, there are no significant trends in 11 out of thirteen of the flights we now discuss this explicitly:- "Alternatively, there may be trends in the particle concentrations that occur during the entire measurement period which spanned around 30

minutes duration. We determine trends in the G_NR and G_R statistics by determining the mean slope (particles / s) during the operating periods; only flights T4 and T6 show trends of greater than 0.1 particles per second (6 particles / minute) when averaged over both G_NR and G_R. Figure 4 shows that there is potentially an increase in the concentrations that are measured during T4, and that there is potentially a bi-modal number concentration measured during the G_NR sampling period for T6. As no trends are evident for the other flights, it can be inferred that there is no evidence of a systematic significant trend in atmospheric concentrations across all flights; any such trends are likely to be random. However, a potential solution to any concern would be to change the three stage sequence from G_NR, G_R, FLY to a five stage sequence of G_NR, G_R, FLY, G_R and G_NR. This sequence is suggested for future investigations."

L285-287 not needed.

This sentence hints at why the performance of the POPS under high windspeed was not as good as that under the low wind speed. We added more text and statistic of flow rate to explain this (L290 – L303).

L301-303. This is a good point. The POPS flow (used to calculate PC in each bin size)

needs to be monitored in each of the different flight positions.

We added the statistics of sample flow rates (Table 6) and more text (L290-L303):

"The variability in the pitch, yaw and altitude of the drone also impacted the orientation of the inlet of the POPS, which ideally should be perpendicular to the horizontal plane. Variations in the orientation of the inlet led to uncertainties in the sample flow rate. Table 6 shows mean sample flow rates with standard deviation at G_NR, G_R, and FLY for all cases. It is clear that for G_NR, the mean flow rates were constant across all tests and the standard deviation in the flow rates were very low. Comparing with G_NR, the mean sample flow rate and the standard deviation were almost unchanged with for G_R. This shows that operating the rotors alone didn't impact the sample flow rate. However, while the mean flow rate during FLY was identical to G_NR, the standard deviations increased during the FLY stage, particularly for the tests under high windspeeds. The mean value of the standard deviation for low windspeed cases was 0.13, while for the high windspeed cases was 0.21 which may influence the accuracy of the POPS measurements."

| | Surface Wind Speed (m/s) | Sample flow rate (cm$^3$/s) | | |
|---|---|---|---|---|
| | | G_NR | G_R | FLY |
| *T1* | 0.5 | 3.04±0.04 | 3.03±0.04 | n/a |
| *T2* | 2.6 | 3.04±0.04 | 3.04±0.05 | 3.03±0.12 |
| T3 | 5.7 | 3.03±0.04 | 3.03±0.06 | 3.03±0.20 |
| T4 | 3.6 | 3.04±0.04 | 3.03±0.05 | 3.03±0.16 |
| T5 | 6.7 | 3.02±0.04 | 3.02±0.04 | 3.00±0.26 |
| *T6* | 1.5 | 3.02±0.04 | 3.03±0.04 | 3.03±0.15 |
| *T7* | 1 | 3.03±0.03 | 3.03±0.05 | 3.03±0.17 |
| *T8* | 4.1 | 3.03±0.03 | 3.03±0.04 | 2.99±0.28 |
| *T9* | 7.7 | 3.03±0.04 | 3.03±0.05 | 3.02±0.21 |
| *T10* | n/a | 3.02±0.04 | 3.02±0.04 | 3.03±0.19 |
| T11 | n/a | 3.02±0.04 | 3.02±0.04 | 3.03±0.22 |
| T12 | n/a | 3.02±0.03 | 3.02±0.04 | 3.04±0.16 |
| T13 | n/a | 3.02±0.04 | 3.03±0.04 | 3.03±0.23 |
| *T14* | n/a | 3.02±0.04 | 3.02±0.05 | 3.00±0.17 |

**Table 6.** Summary of the sample flow rates of each test flight at three stages. n/a = not applicable. The numbers denoted by ±x represent the standard deviation in the sample flow rates during the measurement time period.

L313 – 316. Again what is the flow in each POPS case? Are the counting statics much poorer during FLY than when on the ground?

We have improved our analysis as above.

L331-333 Unfortunately, this point undermines this entire study. If there are real differences that might be confused with instrument performance in flight, the study ideally should address this.

This has been answered in the Overall Comments.

L336-337 This does not make sense. Two additional stages of G_NR are suggested?

Apologies – typo "to a five stage sequence of G_NR, G_R, FLY, G_R and G_NR. This sequence is suggested for future investigations."

L356- end Tailor sweeping claims as suggested.

We don't make sweeping claims – we are very specific about the performance of the UAV under different wind-speed conditions. Please direct us to these sweeping claims if you disagree.

---

## Author Comment (AC2)

**Characterizing the performance of a POPS miniaturized optical particle counter when operated on a quadcopter drone, by Liu et al.**

**Response to Reviewers**

We would like to thank the reviewers for their comments on the manuscript. The reviewers raised some interesting and relevant points that we have addressed in our revised manuscript. Our responses are presented in red.

**Second Referee**

**Overall Comments:**

This manuscript investigated the performance of POPs on a drone and compared with aerosol microphysics measurements observed at the surface. This study provides a series of methods to quantify the uncertainties of the PNCs and PSD measured onboard a drone. The results from this study provide a good reference to the community when one wants to use the POPS or similar instruments to access aerosol profiling issues. The manuscript is well written in English and content structure. My comments are listed bellows:

Higher wind speed increased measurement error. The author explained that it is caused by the drone suffers from variations in pitch and yaw. It is not clear to the reviewer. If the wind direction is fixed during the flight, how the drone changes pitch and yaw frequently. My personal opinion is that the inlet flow rate of POPS is small, the high speed of horizontal wind causes the measurement uncertainty due to the insufficient inlet flow.

The wind direction is approximately fixed during each flight, but the speed is not. Also, due to the limitation of the power and control system, the drone is hard to keep stable under any violent gusts. But we agree the reviewer's opinion that the inlet flow rate cause uncertainties of the POPS. We added the table and text as follow (L290 – L303):

"The variability in the pitch, yaw and altitude of the drone also impacted the orientation of the inlet of the POPS, which ideally should be perpendicular to the horizontal plane. Variations in the orientation of the inlet led to uncertainties in the sample flow rate. Table 6 shows mean sample flow rates with standard deviation at G_NR, G_R, and FLY for all cases. It is clear that for G_NR, the mean flow rates were constant across all tests and the standard deviation in the flow rates were very low. Comparing with G_NR, the mean sample flow rate and the standard deviation were almost unchanged with for G_R. This shows that operating the rotors alone didn't impact the sample flow rate. However, while the mean flow rate during FLY was identical to G_NR, the standard deviations increased during the FLY stage, particularly for the tests under high windspeeds. The mean value of the standard deviation for low windspeed cases was 0.13, while for the high windspeed cases was 0.21 which may influence the accuracy of the POPS measurements."

We include the following table showing the mean and standard deviation of the sample flow rate

| | Surface Wind Speed (m/s) | Sample flow rate (cm$^3$/s) | | |
|---|---|---|---|---|
| | | G_NR | G_R | FLY |
| *T1* | 0.5 | 3.04±0.04 | 3.03±0.04 | n/a |
| *T2* | 2.6 | 3.04±0.04 | 3.04±0.05 | 3.03±0.12 |
| T3 | 5.7 | 3.03±0.04 | 3.03±0.06 | 3.03±0.20 |
| T4 | 3.6 | 3.04±0.04 | 3.03±0.05 | 3.03±0.16 |
| T5 | 6.7 | 3.02±0.04 | 3.02±0.04 | 3.00±0.26 |
| *T6* | 1.5 | 3.02±0.04 | 3.03±0.04 | 3.03±0.15 |
| *T7* | 1 | 3.03±0.03 | 3.03±0.05 | 3.03±0.17 |
| *T8* | 4.1 | 3.03±0.03 | 3.03±0.04 | 2.99±0.28 |
| *T9* | 7.7 | 3.03±0.04 | 3.03±0.05 | 3.02±0.21 |
| *T10* | n/a | 3.02±0.04 | 3.02±0.04 | 3.03±0.19 |
| T11 | n/a | 3.02±0.04 | 3.02±0.04 | 3.03±0.22 |
| T12 | n/a | 3.02±0.03 | 3.02±0.04 | 3.04±0.16 |
| T13 | n/a | 3.02±0.04 | 3.03±0.04 | 3.03±0.23 |
| *T14* | n/a | 3.02±0.04 | 3.02±0.05 | 3.00±0.17 |

**Table 6.** Summary of the sample flow rates of each test flight at three stages. n/a = not applicable. The numbers denoted by ±x represent the standard deviation in the sample flow rates during the measurement time period.

**Specific Comments:**

In the abstract (line 59), the measurement errors induced from turbulence need to be carefully stated since the authors did not provide turbulence measurement data to support the conclusion.

Here we would like to point out that in our new analysis suggest that the impact of rotors are not significant for the performance of the POPS in our tests. The turbulence caused by rotors has been observed by previous studies (Wen et al., 2019).

Wen, S., Han, J., Ning, Z., Lan, Y., Yin, X., Zhang, J., Ge, Y., 2019. Numerical analysis and validation of spray distributions disturbed by quad-rotor drone wake at different flight speeds. Computers and Electronics in Agriculture 166, 105036.

Introduction: The first paragraph is too long and can be divided into three paragraphs.

We have shortened the instruction and separated it into three paragraphs.

Line 138: please check latlon format.

We rephrase the sentence as following:

"As part of the CLARIFY-2017 and LASIC campaign, the POPS was deployed at the ARM mobile site on Ascension Island located in the mid-Atlantic (7.96° S, 14.37° W) alongside an ARM operated SMPS."

Lines 157-160: not clear to reviewer, rewritten the sentence is needed.

This sentence was superfluous and we have removed it.

Line 278: T14 was marked as red in the table.

This mistake in Table 4 has been corrected.

Lines 250 and 351. The values from what kind of surface measurements should be specified.

We added text as following:

"The close to surface data were collected by the UAV-mounted POPS when the drone was 1-3 meters above the surface."

In Figure 3. The unusual spike values (bad data) may remove from the figure. X-axis label should be improved.

We would like to keep the spikes because they are not bad data. These peaks show the calibration of the instrument. Instead we explain this in the caption:-

"Spikes in the CO data occur at the beginning of each day when the instrument is in calibration mode"

In Figure 5: The dark blue line can be changed as green.

We would like to keep the color because they are friendlier for color blind people.

Table 5: All flights à all cases. The PNC MAD (%) might merge to the same raw of PNC RMSD(%).

Thanks for the suggestion. We choses to keep the MAD and RMSD separate because it provides a clearer analysis of where the errors originate.

Table 6 might present in two tables, same as tables 4&5.

We could, but we prefer to keep them as a single table (now Table 7) to keep the number of tables more manageable. We think that the information presented in the Table is clear enough.

---

## Author Response (AR2)

**Characterizing the performance of a POPS miniaturized optical particle counter when operated on a quadcopter drone, by Liu et al.**

**Response to Associate Editor**

We would like to thank the associate editor for his comments on the manuscript. The associate editor raised some interesting and relevant points that we have addressed in our revised manuscript. Our responses are presented in red.

**Overall and Specific Comments:**

I would like to thank the authors for their efforts in addressing the comments on their originally submitted manuscript that were made by the peer reviewers. I think the manuscript presents a reasonable analysis of the data the authors have collected on the performance of their POPS instrument in both a ground comparison with a SMPS reference instrument during the CLARIFY 2017 campaign and by comparing flight (hover) to non-flight (ground) performance when mounted on a quadcopter UAS.

I would encourage the authors to address a few additional comments prior to publication:

L40: perhaps "operating on the ground and when hovering at 10 m." since both are with the POPS mounted on the UAS.

We modified the text to "We report the root mean square difference (RMSD) and mean absolute difference (MAD) in particle number concentrations (PNCs) when mounted on the UAV and operating on the ground and when hovering at 10m."

L60: "acting as condensation nuclei" -> "to act as cloud condensation nuclei"

Agreed.

L109: "use the" -> "used a" or "included a"

Agreed.

L138: SMPS sample was dried while POPS was not. How close was the POPS temperature to the ambient sampling temperature?

Unfortunately, the POPS doesn't have an inside temperature sensor within the optical cavity. The POPS only has a temperature sensor on the outside on the Beagleboard computer to provide the ambient temperature.

L150: it would be useful for readers if the authors would add some detail to the process used

to adjust the sizing (signal -> diameter) for the change in refractive index from PSL to that assumed for biomass burning. Was the Mie calculation (polarized?) carried out for the POPS light collection geometry or more generally? Mie code used for calculation?

The Mie calculations were made for polarized light and for the geometry specific to the POPS. We have added wording to this section (L148-L150) and also added a detailed description of the process as Appendix A. We feel that this will serve as a useful reference document for future operations.

L161: SMPS range is stated as 0.01 µm to 1.00 µm, but in Fig 2 and the description of the overlap size range the upper limit is 0.45 µm.

Apologies – this was a typo – we have now corrected this to "while the SMPS covers diameter ranging from around 0.01 to 0.45µm."

L180 (Table 1): Table 1 as constructed seems unnecessary. The dates are already included in Table 3 along with other important information—perhaps Table 1 could be replaced by Table 3 with an additional column for time.

Agreed. Actually, we have also incorporated the information on the profiles (previous Table 2) into a new Table 1 as this seemed most efficient. Because AMT cannot display tables in colour, we also give up the colour of the test numbers in Table 2, Table 4, and Table 5, but keep the different font.

L209: "contribute to 93% of the scattering" should be just "contribute 93% of the scattering"

Agreed.

L211-: it would be useful to include in the text some of the details that appeared in your response to reviewer 1's question about the CLARIFY PCASP data that are shown in Fig 2: POPS and SMPS were sampling at 330 m while the BAE146 PCASP measurements were 1.9-7.3 km and the potential causes of the divergence at larger Dp.

We added the statement (L205-207): "The POPS and SMPS were sampling at around 330m altitude ASL when at the LASIC site while the PCASP data from the CLARIFY campaign was collected from 1.9-7.3km ASL"

L239: "mass mixing ratio"—Figure indicate volume mixing ratio (ppmv)

Agreed. Amended.

L251->: it seems that the typical (although not universal) increase in PNC between G_R and G_NR could include a contribution from suspension of particulate material by the near-ground prop wash (also coarse more impacted than accumulation?), but the same is not

true for the even larger typical increase between FLY and G_NR. What is the proposed mechanism by which higher wind speeds lead to artificially higher PNCs in flight?

See below

L293: "Variations in the orientation of the inlet led to uncertainties in the sample flow rate." Is vague. The POPS flow and uncertainty in the flow are independent of instrument orientation. One might postulate based on the observed increase in the scatter of the reported flow (whether real or perceived) during flight (FLY relative to G_R), that the fast adjustment of the platform to changes in wind produces this variability in the flow. The noise in the flow does not increase appreciably from G_NR to G_R, but there is an observed mean increase in PNC.

See below

L302: How (though what mechanism) do you expect the noise in the reported flow impacted the PNC measurement over time? L299 states that the mean flow rate was unchanged, but the result seems to be a systematic increase in the measured PNC.

The three comments above are rather inter-linked so we deal with all here.

We agree that prop wash could potentially increase the suspension of particles when operating on the ground. It is difficult to establish definitive reasons for the larger PNCs during FLY when compared to G_NR, and they may not be artificial. One reason for FLY registering higher PNCs might be that the surface acts as a sink for particles via dry deposition, the rate of which will primarily dependent upon the friction velocity of the surface which will depend on the details of the fetch, i.e. the direction of the wind (e.g. Pellerin et al., 2017). One recent measurement campaign using a miniaturized OPC (the LOAC) uses a tourist balloon tethered in Paris to examine the PNC within the lowest 50m of the atmosphere (Renard et al., 2020), but the vertical resolution of the measurements that are reported are insufficient for us to determine whether they find similar results over the lowest 10m of the atmosphere. Far more attention has been given to the vertical profile of ozone owing to the direct health impacts of ozone and the impacts of ozone and vegetation damage (Clifton et al., 2020).

However, as suggested by the editor, these impacts may also be artifacts: a potential mechanism could be that strong winds causes the attitude of drone to change, which changes the angle of the inlet relative to the horizontal plane horizontal plane which changes the sample flow rate.

We therefore add a short paragraph at line 396 – in the discussion of the results rather than in the results themselves as this is a more logical place to include the discussion: "While the increase in PNC from G_NR to G_R might be explained by generation or resuspension of aerosols from the surface by the rotors of the UAV, the increase from G_R to FLY is more difficult to attribute. The surface acts as a net sink in aerosols through dry deposition which

could lead to an increase in PNC with altitude (e.g. Pellerin et al., 2017), but there are confounding factors from changes in the attitude of the drone and rapid changes in the attitude necessary for stabilizing the position of the UAV during FLY that could also influence the measurements. Indeed, there is evidence that fast adjustments to the attitude of the UAV increase the variability in the flow rate reported by the POPS sensor, particularly at higher windspeeds, where these corrections are larger."

Pellerin, G., Maro, D., Damay, P., Gehin, E., Connan, O., Laguionie, P., Hébert, D., Solier, L., Boulaud, D., Lamaud, E. and Charrier, X., 2017. Aerosol particle dry deposition velocity above natural surfaces: quantification according to the particles diameter. *Journal of Aerosol Science*, *114*, pp.107-117.

Renard, J.B., Michoud, V. and Giacomoni, J., 2020. Vertical profiles of pollution particle concentrations in the boundary layer above Paris (France) from the optical aerosol counter loac onboard a touristic balloon. *Sensors*, *20*(4), p.1111.

Clifton, O.E., Fiore, A.M., Massman, W.J., Baublitz, C.B., Coyle, M., Emberson, L., Fares, S., Farmer, D.K., Gentine, P., Gerosa, G. and Guenther, A.B., 2020. Dry deposition of ozone over land: processes, measurement, and modeling. *Reviews of Geophysics*, *58*(1), p.e2019RG000670.

L309: "differences at sub-micron sizes are less than those at" or "differences in sub-micron aerosol are smaller than those in super-micron aerosol"

We modified the text to: "It also shows that the differences in sub-micron sizes are less than those in super-micron sizes at G_R and FLY."

L312: "summarizes the PSD percentage differences for the two modes"

We modified the text to: "Table 5 summarizes the PSDs percentage differences for two modes at G_R and FLY for each case."

L327: What is meant by POPS "operated in" accumulation mode vs coarse mode? Are these not just different parts of the size distribution (per line 311)?

We modified the text to (L328-L331): "Generally speaking, RMSDs and MADs indicate the impact of rotors and UAV attitude on the performance of the POPS in measuring the accumulation mode is lower than when in measuring the coarse mode, for all cases."

L391: What would the hypothetical mechanism for the rotor induced increase in coarse mode aerosol be?

We now state (L396): "While the increase in PNC from G_NR to G_R might be explained by generation or resuspension of aerosols from the surface by the rotors of the UAV, the

increase from G_R to G_FLY is more difficult to attribute"

Figure 3 caption: perhaps "(a) Time series of SMPS and POPS particle concentrations in the diameter range 120 – 450 nm measured during the LASIC/CLARIFY-2017 campaign. (b) Ratio of the POPS to SMPS concentrations shown in (a)." I agree with the spikes in the CO trace from the daily calibrations is distracting. In panel (e), it would be better to not include the lines between gaps in the AOD data.

We change the caption to: "(a) Time series of SMPS and POPS particle concentrations in the diameter range 120 – 450 nm measured during the LASIC/CLARIFY-2017 campaign. (b) Ratio of the POPS to SMPS concentrations shown in (a)."

We have also removed the calibration spikes from the CO plot, and the dashed lines from the AOD plot.

[Figure]

**Figure 3.** From top to bottom. (a) Time series of SMPS and POPS particle concentrations in the diameter range 120 – 450 nm measured during the LASIC/CLARIFY-2017 campaign. (b) Ratio of the POPS to SMPS concentrations shown in (a). (c) Geometric mean diameter from SMPS. (d)

Carbon monoxide mixing ratio from Los Gatos Research CO analyser, and (e) AOD from Cimel sun- photometer.

Figure 4: it would seem that using a consistent binning across the G_NR, G_R and FLY PDFs of each flight would provide a better comparison of the respective PNC distributions than the variable bin widths that are presented.

Thank you for the suggestion. We have modified the Figure 4 that using a consistent binning across the G_NR, G_R, and FLY PDFs of each flight.

[Figure]

**Figure 4.** Probability density functions (PDFs) of PNCs in each case. A constant bin width is utilized

across the G_NR, G_R, and FLY PDFs of each flight.